# Spatial Pruned Sparse Convolution for Efficient 3D Object Detection

Jianhui Liu[1]* Yukang Chen[2]* Xiaoqing Ye[3] Zhuotao Tian[2] Xiao Tan[3] Xiaojuan Qi[1]†

[1]The University of Hong Kong [2]The Chinese University of Hong Kong [3]Baidu

## Abstract

3D scenes are dominated by a large number of background points, which is redundant for the detection task that mainly needs to focus on foreground objects. In this paper, we analyze major components of existing sparse 3D CNNs and find that 3D CNNs ignore the redundancy of data and further amplify it in the down-sampling process, which brings a huge amount of extra and unnecessary computational overhead. Inspired by this, we propose a new convolution operator named spatial pruned sparse convolution (SPS-Conv), which includes two variants, spatial pruned submanifold sparse convolution (SPSS-Conv) and spatial pruned regular sparse convolution (SPRS-Conv), both of which are based on the idea of dynamically determining crucial areas for redundancy reduction. We validate that the magnitude can serve as important cues to determine crucial areas which get rid of the extra computations of learning-based methods. The proposed modules can easily be incorporated into existing sparse 3D CNNs without extra architectural modifications. Extensive experiments on the KITTI, Waymo and nuScenes datasets demonstrate that our method can achieve more than 50% reduction in GFLOPs without compromising the performance. Code and models are available at this link.

## 1 Introduction

3D object detection has always been a research field of great interest due to its wide applications in autonomous driving, virtual reality, and robotics. However, compared with the 2D detection task, LiDAR-based 3D detection is more challenging due to the inherent characteristics of LiDAR points such as disorders, irregularity, and non-uniformity. Recently, voxel-based methods [23, 10, 2, 37, 31, 18] with deep 3D sparse convolutional neural networks (CNNs) have become one of the major research streams to tackle 3D object detection problem due to its effectiveness and simplicity. Nevertheless, the improved accuracy is often accompanied by increased computational costs [30, 36, 3], limiting its applicability in practical systems. This motivates us to investigate potential redundancies in the detection model that can be safely avoided to improve efficiency without sacrificing accuracy.

When delving into the task of 3D detection, we found that 3D data itself has high redundancy as shown in Fig. 1(a). Notably, compared with background points in each stage of the sparse CNN, the proportion of foreground points in the entire scene is extremely low (around 5%), which shows that less-informative background points occupy the major areas of a scene. However, the existing 3D sparse CNNs are applied uniformly to the whole scene, causing a considerable amount of computation on the background areas, which might be potentially redundant. Intuitively, if such areas can be identified and selectively skipped, the computational costs have the potential to be dramatically reduced without deteriorating the performance.

Besides the redundancy of the data, the model design itself also brings redundancy. To avoid aggressive down-sampling, 3D sparse CNNs adopt a dilation-like design when applied to perform

---

*Equal Contribution. † Corresponding Author.

36th Conference on Neural Information Processing Systems (NeurIPS 2022).

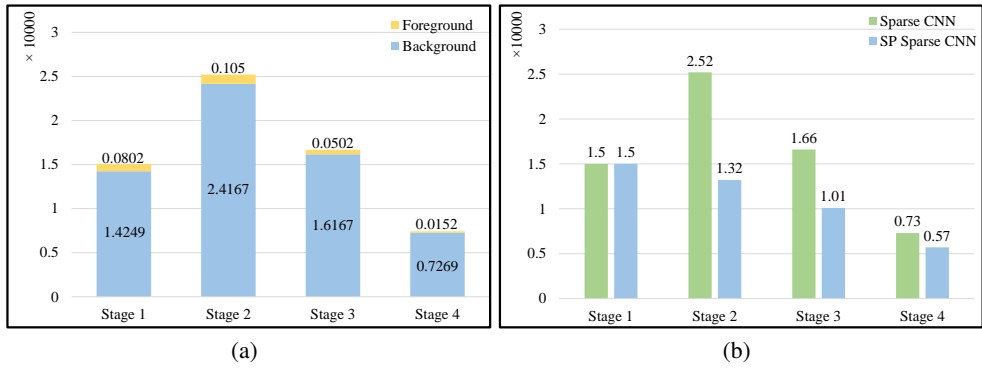

Figure 1: (a) Comparison of the foreground and background ratios. (b) By replacing SPRS-Conv with its counterpart in sparse CNN, unnecessary positions are effectively suppressed from being activated.

the down-sampling operation. Specifically, it will compute features for adjacent empty voxels as shown in Fig. 2 (b): the yellow voxels which are initially empty will be assigned with computed features. Consequently, after convolutional (stride > 1) down-sampling, the number of non-empty voxels might be increased rather than decreased as shown in Fig. 1 (b): the number of non-empty voxels even doubled (see stage 2) compared to the input (see stage1). This will undoubtedly increase unnecessary computational costs for follow-up stages.

The above observations prompt us to ask whether there is a way to identify redundant computations that can be pruned to improve efficiency. A solution that has been attempted in the 2D image domain [29, 11] is to add an auxiliary learnable module to predict a soft mask [21, 28] that locates areas to be skipped for computational efficiency. The module often requires additional post-fine-tuning, auxiliary costs for integration, and incurs non-negligible computational overheads.

With the aforementioned considerations, we propose a new simple and efficient sparse convolution operator named Spatial Pruned Sparse Convolution (SPS-Conv), to alleviate redundancies caused by data and model design. The core idea of SPS-Conv is to find redundancy in the model dynamically. To avoid the overheads of learning-based methods for simplicity and efficiency, we investigate whether the features from the model itself contain useful cues that can be leveraged to identify redundancies. Fortunately, we find that the magnitude of features could be a robust signal to reflect the importance where locations with smaller magnitudes are more likely to be redundant areas. This leads us to a magnitude-based sampling module which is incorporated into 3D convolution layers to reduce redundancy in data and model.

Specifically, to address data redundancy, the submanifold variant of SPS-Conv, termed as spatial pruned submanifold sparse convolution (SPSS-Conv) can adaptively calculate important positions in the light of the magnitude of features and perform convolutions on these locations, leaving features of redundant locations unchanged. As for model redundancy in the down-sampling process, another variant named spatial pruned regular sparse convolution (SPRS-Conv), which can dynamically determine the position that needs to be inflated. As Tab. 1(b) shows, our method effectively reduces the meaningless expansion caused by convolution (around 50% in stage2). In general, SPS-Conv only keeps the essence and discards the irrelevant ones for pursuing a higher efficiency without compromising performance.

In summary, we propose an efficient convolution operator for spatial redundancy pruning, which can be easily incorporated into existing sparse 3D CNNs for 3D detection. We conduct extensive experiments on both KITTI, Waymo, and nuScenes datasets. The result shows that we can achieve comparable performance with SOTA methods while enjoying 52.4%, 62.46%, and 46.5% GFLOPs reduction.

## 2 Related Work

**Sparse Convolution.** Depending on how it is used, sparse convolution[15] can be subdivided into regular sparse convolution and submanifold sparse convolution. Regular sparse convolution is

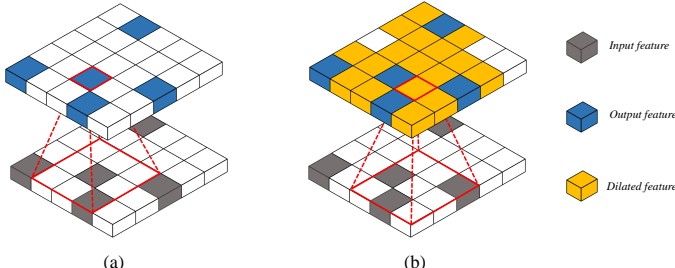

Figure 2: Illustration of conventional sparse convolution. (a) Submanifold sparse convolution (b) Regular sparse convolution.

often used for the down-sample layer, which dilates all input features to its kernel-size neighbors, sacrificing computational efficiency in exchange for a wider interaction of information. In contrast, the submanifold convolution is more like a simplified version of the regular sparse convolution, frequently used in residual blocks[17], only calculates on valid positions, avoiding meaningless expansion and achieving efficient computation.

**Voxel-based Detectors.** In order to process irregular point cloud data, voxel-based methods first convert the point cloud into regular voxel, and then use mature CNNs for feature extraction. However, the computational cost and memory requirement both increase cubically along with the voxel resolution. Thus, it is infeasible to train a voxel-based model with high-resolution inputs. Benjamin et al.[15] proposed a novel convolution operator named submanifold sparse convolution, by reducing calculation on invalid position, which greatly improves the computing efficiency and memory.

Approaches for voxel-based detection methods can be grouped into two categories, *i.e.,* single-stage[12, 16, 34, 35, 39, 40] and two-stage detectors[10, 23, 25, 22, 24]. Single-stage detectors are relatively simple, and directly predict final bounding boxes based on the features extracted by sparse CNN. VoxelNet[41] utilizes a Sparse CNN to extract voxel features from a dense grid. SECOND[31] proposes 3D sparse convolutions to efficiently extract voxel features. HVNet[34] designs a convolutional network that attentively aggregates and projects the multi-scale feature maps to achieve better performance. In contrast, two-stage detectors are more complex but can get higher performance. Part-$A^2$[24] proposed a part-aware and aggregation module to exploit the intra-object part location. PV-RCNN[23] uses key points to extract voxel features for better box refinement.

**Dynamic Kernel Shape Design.** Much literature work on dynamic kernel design for adapting various tasks. Deformable convolution [9] predicts offsets for feature sampling. For 3D scene understanding, KPConv [27] constructs dynamic graph for kernel points. Deformable PV-RCNN [3] applies offset prediction for feature sampling in 3D object detection. Focals Conv [8] learns a dynamic spatially sparsity which enhances the network for spatial modeling capabilities,

## 3 Background and Motivation

In this section, we will introduce the mechanism of sparse convolution, then analyze its redundancy and introduce our motivation.

### 3.1 Preliminary of Sparse Convolution

To better illustrate sparse convolution, we first introduce its general expression. We denote $x_p$ as an input feature with $c_{\text{in}}$ dimension at position $p$, $w \in \mathbb{R}^{K^d \times c_{\text{in}} \times c_{\text{out}}}$ is the weight of convolution kernel, where $d$ and $K^d$ refers to the dimension of the spatial space and spatial size of the kernel respectively. Thus we can formulate the convolution process as follows.

$$y_p \in P_{\text{out}} = \sum_{k \in K^d(p, P_{\text{in}})} w_k \cdot x_{\bar{p}_k}, \text{ where } K^d(p, P_{\text{in}}) = \left\{ k \,|\, p + k \in P_{\text{in}}, k \in K^d \right\}, \quad (1)$$

Here, $P_{\text{in}}$ and $P_{\text{out}}$ refer to the input and output feature space, $k$ is the kernel offset that corresponds to all the valid locations in kernel space $K^d$, $\bar{p}_k = p + k$ denotes all non-empty neighbor voxels around center $p$. Due to the discrete data distribution in 3d space, we define $K^d(p, P_{\text{in}})$ as a subset of $K^d$, leaving out the empty position, which is constrained by position $p$ and input feature space $P_{\text{in}}$.

There are two types of sparse convolutions, namely, regular sparse convolution [14] and submanifold sparse convolution [15], their major difference lies in the output position of the convolution. For regular sparse convolution, as shown in Fig. 2, the position related to the input will be activated during the convolution process. Specifically, these positions are assigned value and combined with the $P_{\text{in}}$ to form a new $P_{\text{out}}$. This process can be easily formulated as

$$P_{\text{out}} = \bigcup_{p \in P_{\text{conv}}} Q\left(p, K^d\right), \text{where } Q\left(p, K^d\right) = \left\{p + k \,|\, k \in K^d\right\}, \tag{2}$$

where $P_{\text{conv}}$ is a subset of $P_{\text{in}}$, representing the position where the convolution needs to be performed. $Q\left(p, K^d\right)$ refers to the position related to $P_{\text{conv}}$ which is restricted by kernel size.

Regular sparse convolution can effectively expand the receptive field like 2D convolution, which is beneficial for 3D sparse data. However, it also brings a computation burden due to the generation of too many active locations, which can lead to a decrease in speed in subsequent convolution layers due to a large number of active points. In order to achieve a better trade-off between efficiency and receptive field, regular sparse convolution is often adopted in the down-sample layer in sparse CNNs.

In contrast, submanifold sparse convolution[15] restricts an output location to be active if and only if the corresponding input location is active, which avoids the dilation in regular sparse convolution, although the receptive field is limited. With a simple modification on Eq. 2, $P_{\text{out}} = P_{\text{in}}$, regular mode can be convert to submanifold mode.

## 3.2 Redundancy Analysis and Motivations

Despite the wide adoption of sparse CNNs in 3D object detection due to their generality and efficiency, spatial redundancy caused by sparse CNNs for this task is still under-explored. Considering the large number of task-independent background points in the 3D scene, a lot of redundant computation will be generated in each stage. In addition, due to the expansion of regular convolution, the background points will become denser, bringing more computational burden to the following stages. Therefore, in response to these problems, we have targeted the improvement of the sparse convolution operator. Details are shown in Sec. 4

## 4 Spatial Pruned Sparse Convolution

In this section, we propose a new efficient sparse convolution operator, named spatial pruned sparse convolution (SPS-Conv). Intending to effectively bring down the unnecessary computation costs caused by the 3D spatial redundancy, two variants of SPS-Conv are investigated, *i.e.,* spatial pruned submanifold sparse convolution (SPSS-Conv) and spatial pruned regular sparse convolution (SPRS-Conv), and both of them are based on the same idea that the redundancy should be dynamically ameliorated in 3D space according to the specific contexts of different individuals. In the following, the magnitude-based sampling strategy is presented in Sec. 4.1, followed by the introduction of SPSS-Conv and SPRS-Conv in Sec. 4.2 and Sec. 4.3 respectively. Then, Sec. 4.4 manifests the generalization ability by showing how our method is applied to generic sparse CNNs.

## 4.1 Magnitude-guided Spatial Sampling

In previous works [29, 11, 5], sampling module is often designed in a learnable manner, *i.e.*, a convolution layer is trained to yield soft masks for sampling, while, at the time of inference, the soft masks are converted to hard ones where elements with lower responses will be simply discarded. Although these methods have been empirically shown effective in 2D tasks, the following issues still exist: 1) Additional supervision is required on the prediction branch, otherwise it is easy to fall into trivial solutions, *i.e.*, the values in the soft mask are all close to 1. 2) The proportion of pruning is hard to be fixed, and it is instead automatically determined throughout the learning process, making it hard to satisfy the specific requirement with certain limited computation budget. 3) Two-stage

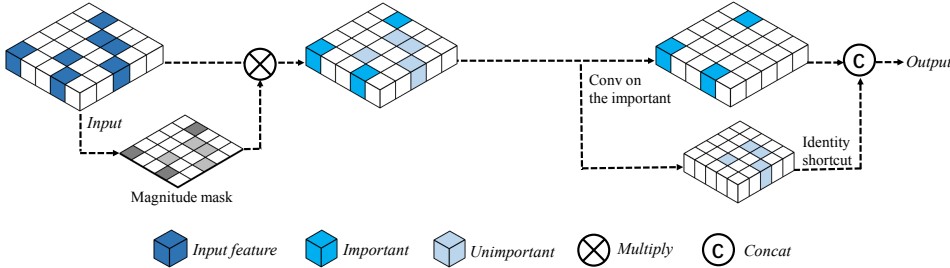

Figure 3: Illustration of magnitude-guided spatial sampling and spatial pruned submanifold sparse convolution (SPSS-Conv). Notably, during the process of SPSS-Conv, "conv on important" refers to performing convolution only at the positions with high magnitude value, and the neighborhood of the convolution still contains the unimportant features.

fine-tuning is required to compensate for the performance loss of converting from soft mask to hard mask, consuming additional time and resources.

Recent literature has shown that the attention mechanism is able to reveal what the model cares about. Following [38, 33], the feature magnitude is adopted for yielding the model's "attention" to highlight the informative elements. For the purpose of getting the magnitude-based attention mask, we first calculate the channel-wise absolute mean values on different voxels, and then the sigmoid function is applied to get the normalized output, which can be formulated as

$$M(F) = \text{Sigmoid}(G(F)), \text{where } G(F) = \frac{1}{C} \cdot \sum_{c=1}^{C} |F_c| , \tag{3}$$

where $F$ and $C$ denote the input feature and its channel. $F_c$ is the feature of the $c$-th dimension. $G$ and $M$ refer to the spatial magnitude map and magnitude mask.

## 4.2 Spatial Pruned Submanifold Sparse Convolution

Considering the fact that the background regions often take the majority in 3D scenes, directly performing the submanifold convolution everywhere of the input volume inevitably involves substantial unnecessary calculations, leading to computational redundancy. Therefore, with a focus on alleviating this issue, we propose the spatial pruned submanifold sparse convolution (SPSS-Conv) that dynamically examines the regions of interest.

Specifically, with the guidance of the magnitude-based attention mask, we separate the elements $P_{\text{in}}$ into two disjoint subsets, i.e., the important set $P_{\text{im}}$ and the unimportant set $P_{\text{nim}}$, where $P_{\text{im}} \cup P_{\text{uim}} = P_{\text{in}}$. We note that various methods can be incorporated for accomplishing the division, such as using a fixed threshold or simply taking the elements with top-k scores. Then we re-weight the input feature map by multiplying it with the magnitude mask, applying submanifold convolution on important positions $P_{\text{im}}$ according to Eq. 1, and concatenate it with the features of unimportant positions $P_{\text{nim}}$ as a residual, as shown in Fig. 3. To this end, the reconstructed output feature can be written as:

$$y_p \in P_{\text{out}} = \begin{cases} \sum_{k \in K^d(p, P_{\text{in}})} (x_{\bar{p}_k} \cdot M(x_{\bar{p}_k})) \cdot w_k, & p \in P_{\text{im}} \\ x_{\bar{p}_k} \cdot M(x_{\bar{p}_k}), & p \in P_{\text{nim}} \end{cases} \tag{4}$$

Since $P_{\text{im}}$ is sparser than $P_{\text{in}}$, convolution enables task-beneficial locations to be highlighted. In contrast, those unimportant positions have relatively small activation values, thus the skip connection can not only reduce the computational overhead but also suppress the diffusion of redundant information.

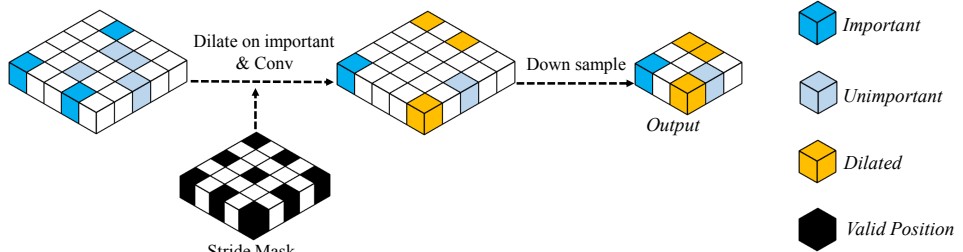

Figure 4: Illustration of spatial pruned regular sparse convolution (SPRS-Conv). The input of SPRS-Conv is generated by magnitude-guided spatial sampling. The figure shows the case of stride=2.

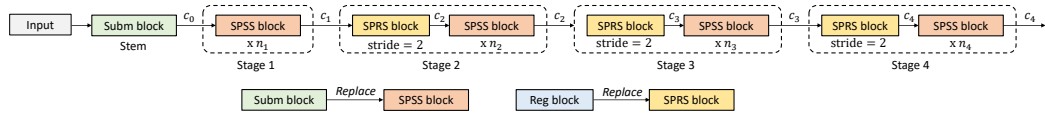

Figure 5: Framework overview. We substitute all the submanifold blocks and regular blocks with SPSS blocks and SPRS blocks, except the block in the stem layer. Each block contains a convolution layer followed by batch normalization and ReLU activation.

## 4.3 Spatial Pruned Regular Sparse Convolution

The "dilation" effect of the regular sparse convolution inevitably causes a large number of adjacent positions to be activated, which brings a greater computational burden to subsequent layers. Recent work [8] shows that the spatially dynamic sparsity in sparse convolution is essential for sophisticated 3D object detection. Motivated by this, SPRS-Conv is designed as a magnitude-based dynamic prediction down-sampling module, which can effectively suppress the amplification effect of spatial redundancy.

Similar to SPSS-Conv, $P_{\text{in}}$ is first divided into two disjoint subsets, *i.e*, $P_{\text{im}}$ and $P_{\text{nim}}$. Unlike the regular sparse convolution that dilates all possible neighborhood locations, leading to sub-optimal efficiency, we instead dynamically choose whether to expand for each position based on the magnitude mask yielded by Eq. 3. Further, we define $P_{\text{dilate}}$, which is a set composed of all $P_{\text{im}}$ positions and the positions within their kernel range. The equation can be written as:

$$P_{\text{dilate}} = \left( \bigcup_{p \in P_{\text{im}}} P\left(p, K^d\right) \right) \cup P_{\text{im}} . \tag{5}$$

Considering the stride during down-sampling, we designed a stride mask $E\left(P\right)$ to ignore those that should be skipped by the stride, which can be formulated as below:

$$E(P) = \left\{ \left( \sum_{i=\{x,y,z\}} \left(p_i \,\%\, s_i\right) \right) = 0 \,\middle|\, p \in P \right\}, \tag{6}$$

where $s_i$ refers to the stride of the convolution on a specific dimension, % denotes mod operation. With $E\left(P\right)$, we can further modify Eq. 2 as:

$$P_{\text{out}} = E\left(P_{\text{dilate}}\right) \cup E\left(P_{\text{nim}}\right) . \tag{7}$$

As shown in Fig. 4, we operate at the specified positions according to Eq. 1 to get the output feature map. Then we down-sample it to the correct size according to the stride. Compared to the regular sparse convolution, our method dynamically prunes $P_{\text{out}}$ conditioned on its specific content. As

| Method | mAP | NDS | Car | Truck | Bus | Trailer | C.V. | Ped | Mot | Byc | T.C. | Bar |
|--------|-----|-----|-----|-------|-----|---------|------|-----|-----|-----|------|-----|
| PointPillars [18] | 30.5 | 45.3 | 68.4 | 23.0 | 28.2 | 23.4 | 4.1 | 59.7 | 27.4 | 1.1 | 30.8 | 38.9 |
| 3DSSD [32] | 42.6 | 56.4 | 81.2 | 47.2 | 61.4 | 30.5 | 12.6 | 70.2 | 36.0 | 8.6 | 31.1 | 47.9 |
| CBGS [42] | 52.8 | 63.3 | 81.1 | 48.5 | 54.9 | 42.9 | 10.5 | 80.1 | 51.5 | 22.3 | 70.9 | 65.7 |
| HotSpotNet [7] | 59.3 | 66.0 | 83.1 | 50.9 | 56.4 | 53.3 | **23.0** | 81.3 | **63.5** | 36.6 | 73.0 | 71.6 |
| CVCNET [6] | 58.2 | 66.6 | 82.6 | 49.5 | 59.4 | 51.1 | 16.2 | 83.0 | 61.8 | **38.8** | 69.7 | 69.7 |
| CenterPoint [37] | 58.0 | 65.5 | **84.6** | 51.0 | 60.2 | 53.2 | 17.5 | 83.4 | 53.7 | 28.7 | 76.7 | 70.9 |
| CenterPoint* | **59.9** | **67.3** | 84.1 | **52.3** | **65.1** | 53.5 | 19.6 | **84.6** | 58.3 | 31.3 | 78.0 | **72.4** |
| SP-CenterPoint | 59.7 | 67.2 | 84.2 | 51.1 | 61.8 | **54.5** | 20.1 | 84.3 | 60.7 | 30.4 | **78.1** | 72.0 |

Table 1: Comparison with other methods on nuScenes *test* split. CenterPoint* denotes our own re-implementation result for CenterPoint.

| Method | mAP | NDS | GFLOPs | Car | Truck | Bus | Trailer | C.V. | Ped | Mot | Byc | T.C. | Bar |
|--------|-----|-----|--------|-----|-------|-----|---------|------|-----|-----|-----|------|-----|
| CenterPoint [37] | 58.6 | 66.2 | 62.9 | 85.0 | 58.2 | 69.5 | 35.7 | 15.5 | 85.3 | 58.8 | 40.9 | 70.0 | 67.1 |
| SP-CenterPoint | 58.5 | 66.1 | **34.3** | 84.9 | 58.1 | 70.2 | 34.7 | 17.2 | 85.3 | 58.7 | 38.6 | 69.7 | 67.0 |

Table 2: Comparison with base method on nuScenes *val* split. GFLOPs only include the floating-point operations in sparse CNNs.

shown in Fig. 1(b), SPRS-Conv greatly reduces the number of voxels in each stage, especially stage2 (about 50%), thus reducing the computational burden caused by background points.

### 4.4 Spatial Pruned Convolution Network

Our method is model-agnostic thus it can be easily plugged into any existing sparse CNNs. A regular frame of sparse CNN is composed of a stem layer and four stages, each of which contains a down-sampling layer and two submanifold sparse convolution blocks. To demonstrate the effectiveness and generalization ability, we replace all regular sparse convolutions and submanifold sparse convolutions in sparse CNN, except those in the stem layer, with our proposed SPRS-Conv and SPSS-Conv, respectively. Detailed experimental results and analysis are presented in Sec. 5.

## 5 Experiments

### 5.1 Experimental Setting

**3D Object Detection Datasets**. We evaluate our method on three challenging benchmarks KITTI [13], Waymo [26] and nuScenes [4]. KITTI dataset consists of 7,481 samples and 7,518 testing samples, where the training samples are generally divided into the *train* split (3,712 samples) and the *val* split (3,769 samples). Waymo dataset contains 1,000 sequences in total, with 798 for training and 202 for validation. As the Waymo dataset is really large-scale, we use $\frac{1}{5}$ data for training. Results in Waymo are evaluated in difficulty LEVEL_1 (L1) and LEVEL_2 (L2) objects. nuScenes dataset is a large-scale autonomous driving dataset, which contains 1,000 driving sequences in total. It is split into 700 scenes for training, 150 scenes for validation, and 150 scenes for testing. It is collected using a 32-beam synced LIDAR and 6 cameras with the complete 360° environment coverage.

**Model Configurations**. *i.e.,* VoxelR-CNN [10], PV-RCNN [23] and SECOND [31] for KITTI, CenterPoint [37] for nuScenes and Waymo. We replace the submanifold convolution and regular convolution in sparse CNNs with SPSS-Conv and SPRS-Conv, respectively, other experimental hyperparameters flowing the default settings of the baseline methods [10, 23, 31, 37, 31].

**Training**. Unlike existing 2D approaches [29, 11, 5] that design the sampling module in a learnable manner which needs a two-stage fine-tune. We train the model in an end-to-end manner. Our SPS-Conv is purely embedded into sparse CNNs without any modifications on network architectures or parameters (*i.e.,* feature dimensions). Our modules contain one hyperparameter, *i.e.,* pruning ratio and we choose to use top-k to divide the pruning part. For spatial pruned submanifold sparse convolution (SPSS-Conv) it refers to the proportion of positions in each stage that can be ignored for calculation, which can be symbolized as $\{r_{s0}, r_{s1}, r_{s2}, r_{s3}\}$. We set it as {0.3, 0.3, 0.3, 0.3 }

| SPSS | SPRS | GFLOPs | Vec_L1 | Vec_L2 | Ped_L1 | Ped_L2 | Cyc_L1 | Cyc_L2 |
|---|---|---|---|---|---|---|---|---|
| | | 76.7 | 72.76 / 72.23 | 64.91 / 64.42 | 74.19 / 67.96 | 66.03 / 60.34 | 71.04 / 69.79 | 68.49 / 67.28 |
| ✓ | | 43.5 | 72.46 / 71.91 | 64.35 / 63.85 | 73.71 / 67.53 | 65.81 / 60.13 | 70.85 / 69.63 | 68.52 / 67.27 |
| | ✓ | 55.2 | 72.80 / 71.96 | 64.36 / 64.07 | 73.99 / 67.84 | 66.05 / 60.40 | 70.42 / 69.69 | 68.85 / 67.37 |
| ✓ | ✓ | 28.8 | 72.68 / 72.13 | 64.66 / 64.16 | 74.39 / 68.02 | 66.00 / 60.36 | 71.08 / 69.84 | 68.47 / 67.28 |

Table 3: Comparison on CenterPoint baseline method on Waymo *val* split, trained with 20% data. GFLOPs only include the floating-point operations in sparse CNNs.

| Method | Easy | Mod. | Hard |
|---|---|---|---|
| Point R-CNN [22] | 88.88 | 78.63 | 77.38 |
| Part-$A^2$ [24] | 89.47 | 79.47 | 78.54 |
| STD [1] | 89.70 | 79.80 | 79.30 |
| SA-SSD [32] | 90.15 | 79.91 | 78.78 |
| PV-RCNN [23] | 89.35 | 83.69 | 78.70 |
| VoTr-TSD [20] | 89.04 | 84.04 | 78.68 |
| Pyramid-PV [19] | 89.37 | 84.38 | 78.84 |
| Voxel R-CNN [10] | 89.41 | 84.52 | 78.93 |
| SP-Voxel R-CNN | 89.69 | 84.57 | 78.89 |

Table 4: Comparison with other methods on KITTI *val* split in AP$_{3D}$ (R11) for Car.

| Method | GFLOPs | Easy | Mod. | Hard |
|---|---|---|---|---|
| Second [31] | 7.6 | 88.36 | 78.75 | 77.57 |
| SP-Second | 3.6 | 88.57 | 78.64 | 77.42 |
| PV-RCNN [23] | 7.6 | 89.35 | 83.69 | 78.70 |
| SP-PV-RCNN | 3.6 | 89.27 | 83.22 | 78.87 |
| Voxel R-CNN [10] | 7.6 | 89.41 | 84.52 | 78.93 |
| SP-Voxel R-CNN | 3.6 | 89.69 | 84.57 | 78.89 |

Table 5: Comparison with base methods on KITTI *val* split in AP$_{3D}$ (R11) for Car. GFLOPs only include the floating-point operations in sparse CNNs.

for nuScenes and {0.5, 0.5, 0.5, 0.5} for Waymo and KITTI. As for spatial pruned regular sparse convolution (SPRS-Conv), the pruning ratio is used for controlling the amount that needs to be inflated when downsampling. We symbolized it as $\{r_{d1}, r_{d2}, r_{d2}\}$, which are set as {0.5, 0.5, 0.5} in nuScenes and Waymo and {0.7, 0.5, 0.3} in KITTI.

## 5.2 Main Results

**nuScenes**. For the sake of fairness, we choose LIDAR-only methods for comparison, model ensembling and additional augmentation are not included during inference. Our method achieves competitive performance on both mAP and NDS on nuScenes *test* split, in Tab. 1. Besides, We apply the method to the base model for a more detailed comparison, as shown in Tab. 2, our method can help the model to preserve the original performance while skipping redundant computation, specifically, SP-CenterPoint's GFLOPs are only 54.5% of the base model.

**KITTI**. We make comparisons with recent state-of-the-art methods on KITTI *val* split. As shown in Tab. 4, Our Spatial Pruned Voxel R-CNN achieves competitive results with other methods. To better demonstrate the effectiveness of our method, we validate our method on popular voxel-based detectors, *i.e.,* Second, PVRCNN, and Voxel-RCNN. Tab. 5 shows that with our method, the GFLOPs of the sparse CNN are reduced by more than 50%, and the matching performance can still be obtained on KITTI *val* split in AP$_{3D}$ in recall 11 positions.

**Waymo**. We present SPS-Conv based on the CenterPoint on the outdoor dataset Waymo [26] in Tab. 3. The results demonstrate the generality of SPS-Conv on large-scale datasets with dense point clouds: SPS-conv still maintains high performance while significantly reducing FLOPs.

| Method | PR. | GFLOPs | mAP | NDS |
|---|---|---|---|---|
| CenterPoint | - | 62.9 | 58.59 | 66.20 |
| | 0.1 | 56.6 | 58.64 | 66.25 |
| | 0.3 | 45.2 | 58.48 | 66.11 |
| SPSS-Conv | 0.5 | 33.5 | 58.19 | 66.08 |
| | 0.7 | 21.6 | 57.80 | 65.69 |
| | 0.9 | 9.7 | 55.21 | 64.23 |

| Method | PR. | GFLOPs | mAP | NDS |
|---|---|---|---|---|
| CenterPoint | - | 62.9 | 58.59 | 66.20 |
| | 0.1 | 59.1 | 58.57 | 66.01 |
| | 0.3 | 55.4 | 58.61 | 66.07 |
| SPRS-Conv | 0.5 | 47.2 | 58.58 | 66.24 |
| | 0.7 | 32.3 | 58.18 | 65.93 |
| | 0.9 | 9.9 | 52.09 | 61.89 |

Table 6: Ablations on pruning ratio in SPSS-Conv on nuScenes *val* split.

Table 7: Ablations on pruning ratio in SPRS-Conv on nuScenes *val* split.

| Method | mAP | NDS | Car | Truck | Bus | Trailer | C.V. | Ped | Mot | Byc | T.C. | Bar |
|---|---|---|---|---|---|---|---|---|---|---|---|---|
| SPSS-Conv | 58.48 | 66.11 | 85.0 | 58.2 | 69.5 | 35.7 | 15.5 | 85.3 | 58.8 | 40.9 | 70.0 | 68.1 |
| SPSS-Conv random drop | 56.02 | 64.72 | 84.4 | 56.3 | 67.3 | 34.3 | 13.8 | 84.1 | 54.3 | 34.9 | 65.2 | 65.6 |
| SPSS-Conv inverse | 55.84 | 64.49 | 84.3 | 56.7 | 68.2 | 33.3 | 14.4 | 83.5 | 54.4 | 34.2 | 63.3 | 66.1 |
| SPRS-Conv | 58.58 | 66.24 | 84.9 | 57.5 | 69.1 | 35.2 | 15.5 | 85.3 | 59.6 | 40.2 | 70.6 | 67.8 |
| SPRS-Conv random drop | 56.58 | 64.77 | 82.7 | 54.8 | 69.4 | 34.1 | 14.9 | 83.6 | 54.0 | 38.6 | 67.5 | 66.0 |
| SPRS-Conv inverse | 16.72 | 39.29 | 29.6 | 9.4 | 16.3 | 1.7 | 0.6 | 24.2 | 11.8 | 3.6 | 29.0 | 41.1 |

Table 8: Ablations on the importance of position with high magnitude on nuScenes *val* split.

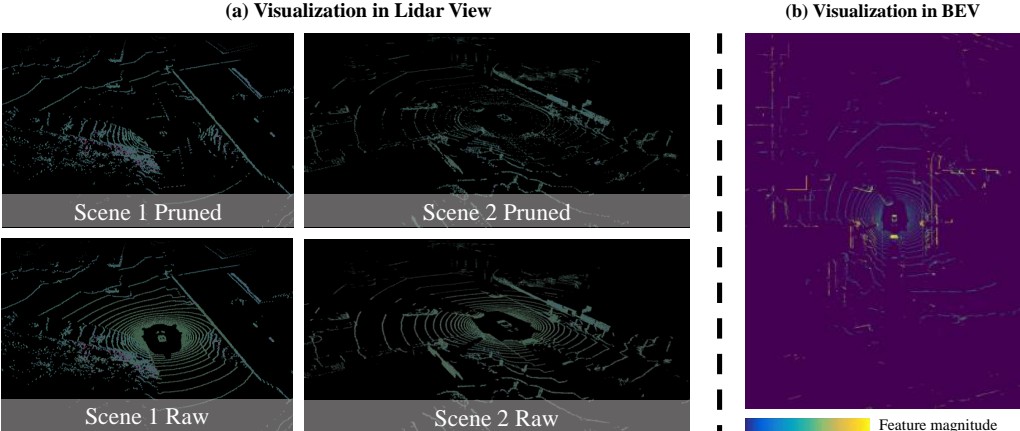

Figure 6: Visualization of SPS-Conv in lidar view and BEV. (a) Comparison before and after pruning in lidar view. (b) Visualization of magnitude in BEV.

## 5.3 Ablation Study

**SPS-Conv Pruning Ratio**. The pruning ratio (PR.) of the SPS-Conv is used to control the proportion of unimportant positions selected in each block, the higher the proportion, the fewer positions are involved in the calculation. We evaluate on nuScenes *val* split upon CenterPoint. As shown in Tab. 6 and Tab. 7, there is a relatively obvious and proportional decrease in GFLOPs, in contrast, the performance drop is not so severe, except when the pruning ratio increases to 0.9. This also reveals that 3D Scenes contain spatial redundancy, and when we selectively skip these redundant positions, it will not affect the performance.

**Sampling and Interpolation Strategy in SPSS-Conv**.
In order to verify the effectiveness of our method, we conduct a combination of several sampling and interpolation methods, where the pruning ratio is limited to 0.5. Results are shown in Tab. 9, it is not difficult to find that the learnable method and magnitude method have comparable results which makes us think that magnitude is another manifestation of network attention, so we select important regions according to this logic, which conforms to the inherent performance of the network. In addition, for the interpolation operation, compared to

| Method | mAP | NDS |
|---|---|---|
| magnitude + skip | 58.19 | 66.08 |
| magnitude + avg pooling | 58.03 | 65.99 |
| conv3x3 + skip | 57.29 | 65.37 |
| conv3x3 + avg pooling | 57.57 | 65.61 |
| linear + avg pooling | 57.67 | 65.54 |

Table 9: Ablations on sampling and interpolation in SPSS-Conv on nuScenes *val* split.

the average pooling operation, a simple skip connection can play a good role. A possible explanation is that these predicted unimportant regions are not task-essential, so even using the original values would not have much impact on the results.

**Importance of Position with High Magnitude**. To verify that the location chosen by this mechanism is task-friendly, we exchanged the calculation method of positions with high magnitude and other positions. As shown in Tab. 8, for SPSS-Conv, it can be seen that when we reverse the important position, there is considerable performance drop in all categories. This phenomenon is more obvious in small objects, especially for the traffic cone and bicycle categories (even reaching about a 7% performance drop). Large object categories (such as vehicles) with more points are less sensitive to the sampling method but still have a notable performance drop. Compared with the experimental

results of SPSS-Conv, the inversion experiment of SPRS has a more exaggerated performance loss. When inversion is not performed, SPRS will suppress the irrelevant features with low magnitude which reduced in the downsample part, showing that there is no obvious performance loss in the result. However, when we choose to suppress positions with large magnitudes, the spatial redundancy is amplified, and important features cannot be effectively expanded, and even discarded due to downsampling. After the above features are converted to Bird's Eye View (BEV), since the number of foreground points from the input is weakened, the effective features extracted by BEVbackone are very limited, resulting in a great degree of performance degradation.

**Magnitude-base Pruning vs Random Drop**. Compared to magnitude-based pruning, we observe using random drop as an indicator will lead to a certain loss in performance (around 2%), shown in Tab. 8. This is caused by the randomness, part of the foreground is discarded, resulting in performance degradation. However, the important part still has a high probability of being selected, which also guarantees performance to a certain extent. Despite its degraded performance, the random drop method also has a certain degree of randomness. This is not desirable in practical applications as it may have the chance to lose some safety-critical areas which will cause problems in safety-critical applications.

### 5.4 Visualization

**Visual Analysis**. To better understand points pruned by our magnitude criterion, we visualize point clouds before and after pruning. The comparison results are shown in Fig. 6(a), the original images and the pruned images are provided respectively. We observe that most of the foreground points are preserved. For the background areas, points that fall in vertical structures, such as light, poles, and trees, are also preserved as they tend to be hard negatives, and easily confused with foreground objects. These points require a deep neural network with a certain capability to process in order to recognize them as background. In contrast, background points in flat structures such as road points are largely removed because they are easily identifiable redundant points.

**Why foreground points with high feature magnitude?** We visualize magnitude in BEV for a clearer comparison, as shown in Fig. 6(b), It is not difficult to see that the feature magnitude of most foreground points is relatively large (vehicles). In contrast, the simple background points (ground) show in small magnitude, which are easier to be distinguished by the network. To gain more insights into why high feature magnitude corresponds to the above patterns, we conjecture that this is caused by the training objective in 3D object detection. When training a 3D object detection model, the focal loss is adopted as default in 3D object detection. When we look closer at the focal loss, it will incur a loss on positive samples and hard negatives while easy negatives are removed from the loss. Thus, this will generate gradients in the direction that can incur an update of features for areas with positive samples and hard negatives. This can eventually make a difference in their feature magnitudes in comparison with areas for easy negatives which are less frequently considered in the optimization objective.

## 6   Concluding Remarks

In this paper, we investigate the spatial redundancy in sparse CNNs and propose a new simple and efficient convolution operator SPSS-Conv with two variants, *i.e.,* SPSS-ConV and SPRS-Conv. Extensive experiments have proved the following conclusions: 1) Magnitude can serve as a good indicator for sparse CNNs to dynamically identify the informative points in diverse scenes. 2) The overwhelming background region in the 3D scene results in spatial redundancy, which can be pruned by our method without adversely affecting the performance. 3) Selectively inflating the elements near the region of interest during the down-sampling process not only saves computation but also keeps the necessary information intact. 4) The proposed SPSS-Conv can be effortlessly applied to generic sparse CNNs without specific structural constraints. We hope our work can provide new thoughts for inspiring future research in the community.

## 7   Acknowledgement

This work has been supported by Hong Kong Research Grant Council - Early Career Scheme (Grant No. 27209621), HKU Startup Fund, and HKU Seed Fund for Basic Research.

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
