# OpenReview forum: "Spatial Pruned Sparse Convolution for Efficient 3D Object Detection"
_NeurIPS.cc/2022/Conference — NeurIPS 2022 Accept_

### Official Review · Reviewer_aQxF · 2022-07-10

**Rating:** 5
**Confidence:** 5
**Soundness:** 3 good
**Presentation:** 3 good
**Contribution:** 3 good

**Summary:**

In this paper, the authors study a previously under-explored direction in 3D deep learning acceleration: spatial redundancy. The major observation from the authors is that most points in the large scale LiDAR scans are background points. Thus, it is possible to reduce the computation on these points. The authors then design spatial pruned (submanifold/regular) sparseconv layers using magnitude-based importance masks. Experiment results show 2x FLOPs reduction without accuracy loss.

**Questions:**

Please address my comments in "Weaknesses".

**Limitations:**

See my comments in "Weaknesses". The major concern is on latency reduction and performance on larger scale datasets such as Waymo.

**Strengths And Weaknesses:**

Strengths:

- As I have mentioned in the summary, this work explores spatial redundancy in 3D point clouds, a new direction in efficient 3D deep learning.
- The paper is clearly written and most of the figures look good to me.
- Experiment results are promising: 2x FLOPs reduction with no accuracy loss on nuScenes. Note that the baseline is a representative network, CenterPoint.

Weaknesses:

- The biggest question related to this work is whether such reduction in FLOPs can be translated to latency improvement. Especially when you consider latest sparse conv implementations in spconv 2.x which uses highly customized computation kernels, I'm a little bit concerned whether such pruning will lead to load imbalance between different CUDA threads and limit the speedup. Also, I'm curious what is the overhead introduced to calculate the importance masks.
- It will be great if the authors can add more experiments on larger-scale datasets, such as Waymo. Waymo is much denser than nuScenes, it will be interesting to see the compression ratio and speedup. If the paper is eventually accepted, I'm also interested in seeing some application in sensor fusion methods on nuScenes.
- [Minor] I would suggest the authors to make Section 3 more compact or move some details to the appendix as the target readers can be very familiar with sparse conv.

---

> ### Author Response · Authors · 2022-08-02
> **Response to Reviewer aQxF (2/2)**
>
> **Q4:  Performance, Compression ratio, and speedup on Waymo dataset?**
>
> **A4:**  In this part, we evaluate our model on the Waymo dataset. Due to storage reasons, all experiments kept the batch size as 1 and tested on a single A100 GPU.
>
> We report the performance (table in commen response Q2),  speed, and FLOPs on the Waymo dataset in the following Table.  Our method can effectively reduce GFLOPs (around 63%). Although, FLOPs cannot all translate into speed improvements. But we still have a nearly 20% speedup in latency reduction due to implementation optimization and hardware issues discussed in (Common respose Q1). We believe we still have a room for optimization to further improve the efficiency by implementing customized CUDA fuctions.
>
> | Method / speed(ms) | Waymo (VoxelResNet) | speed up | GFLOPs |
> | ------------------ | ------------------- | -------- | ------ |
> | baseline           | 37 ms               | None     | 76.7   |
> | spss               | 32 ms               | 13.5%    | 43.5   |
> | sprs               | 33 ms               | 11%      | 55.2   |
> | sprs+spss          | 30 ms               | 19%      | 28.8   |
>
> **Q5: The details of spconv in section 3 are somewhat redundant ?**
>
> **A5:** Thanks for the suggestion. We will put some details of spconv in the appendix according to your suggestion, and add Waymo and latency reduction results to the main text.

---

> ### Author Response · Authors · 2022-08-02
> **Response to Reviewer aQxF (1/2)**
>
> **Q1：The latency improvement of the SPS-Conv?**
>
> **A1**:   Please refer to the Q1 in the Common question section. In the table, we show the latency reduction of SPS-Conv. Among them, SPS-Conv and SPRS can get a speed increase of more than 10% when used alone, and they can deliver a higher speed increase (around 20%) when used together.  Note that our current implementations are based on *Pytorch functions* and *spconv 2.x* without implementation engineering. We believe that customized CUDA implementation will further help reduce the running time as the running time is closely also related to implementation besides the model complexity.
>
> We will continue to optimize this code to make it more efficient.  We are sorry that we cannot provide this because of the short rebuttal period. We commit to open source it as a variant of spconv to benefit the community.
>
> **Q2: The overhead introduced to caculate the important masks ?**
>
> **A2:**  Thanks for the suggestion. Following the suggestion, we count the time it takes to generate the mask in the convolution. The results are as follows:
>
> | Method / speed(ms) | KITTI (VoxelNet) | KITTI (mask time) | nuScenes (VoxelResNet) | nuScenes (mask time) |
> | ------------------ | ---------------- | ----------------- | ---------------------- | -------------------- |
> | spss topk          | 36 ms            | 1.7 ms            | 44 ms                  | 4.6 ms               |
> | sprs topk          | 33 ms            | 0.4 ms            | 44 ms                  | 0.9 ms               |
>
> **Impact on latency：** It should be mentioned that the generation of masks is based on torch.argsort(). Since PyTorch optimizations are not ideal, this part does generate additional time consumption. And this effect is more pronounced as the number of points increases. At present, the time consumption generated by the mask is still within an acceptable range as shown in the table.  We will use the divide and conquer algorithm to write a customized CUDA module to accelerate topk operation, which would further improve the latency.  We are sorry for not being able to do this constrained by the short time of rebuttal. *Note that our model still obtains around a 20% overall reduction in latency even with this naive implementation without sacrificing accuracy.*
>
> **Q3：Whether such pruning will lead to load imblance between different CUDA threads and limit the speed up ?**
>
> **A3**:  Thanks for your valuable comments. The calculation of Spconv is mainly divided into two parts (1) generating the index pair and (2) general matrix multiplication (GEMM). We analyze these two aspects separately:
>
> First of all, for the generation of index pairs, we implement it by constraining the output position based on the index mask. Specifically, we only need to pass the index mask into the kernel function as a parameter and use a rule to determine whether the original index pair satisfies the constraints of the index mask. We believe that this part does not account for a high proportion of the overall network inference time, as shown in the table in A2, the impact on CUDA threads thus can be ignored.
>
> Secondly, For GEMM, the implementation of spconv is calculated along the spatial dimensions of the kernel, eg. kernel size: 3x3x3. Different spatial locations are calculated at different iterations and will not affect each other. You might have the impression that there exists a large difference in terms of the number of points at different spatial locations, causing an imbalance in computation. However,  we argue that this again will not lead to load imbalance between CUDA threads because different spatial positions are mapped to independent GEMMs and each GEMM is performed in a dense manner.

---

> ### Comment · Reviewer_aQxF · 2022-08-05
> **Thanks for the response.**
>
> Thank you for your reply which resolves most of my concerns, I will keep my score.
>
> Note: For Q3, I am talking about the implicit GEMM implementation.

---

> > ### Author Response · Authors · 2022-08-06
> > **Author Response**
> >
> > We sincerely thank the reviewer for the constructive feedback and support.

---

### Official Review · Reviewer_B4Lc · 2022-07-11

**Rating:** 6
**Confidence:** 4
**Soundness:** 3 good
**Presentation:** 4 excellent
**Contribution:** 4 excellent

**Summary:**

This paper presents a method for dynamically determining which features on which to perform convolutions in a sparse convolutional network. In particular, the authors apply a sigmoid on top of the magnitude of each feature to determine a score, on which they threshold to determine active sites in each feature map. The motivation is that features with higher magnitude are more important to the network. In the forward pass, only features with a score above a set threshold have convolutions applied, while low score features are passed through via a skip connection. Features are also weighted by this score. Experiments are performed by replacing of a regular sparse convolutional network with the proposed dynamic sparse convolution blocks, where minimal performance regressions are seen despite significant reductions in GFLOPs. Ablations are also provided demonstrating that large score thresholds can be used with minimal impact on performance.

**Questions:**

Please see the strengths and weaknesses section. My main concern regards the ablations in Table 6, which would be alleviated with more explanations/exploration of the underlying assumption of this work.

**Limitations:**

No limitations section is provided. Some discussion of the validity of the base assumption would help future readers.

**Strengths And Weaknesses:**

The authors provide a relatively simple solution to the dynamic sparse convolution problem, given the intuition that higher feature norms correlate with more important features. The method can be dropped into any existing sparse convolution network, and is demonstrated to have significant reductions in computation cost. The sparsity can also be controlled at test time by tuning the score threshold. The method is well explained with nice figures outlining the pipeline, and the results are compelling, with good coverage across two datasets.

Firstly, the intuition that features with higher norms are more important mostly makes sense, but some experimental validation would be very helpful. In particular, as the convolution operator applies the dot product between features, it's possible that a feature has relatively low magnitude compared to others, but high magnitude in a channel that also has high magnitude in the convolutional kernel. This may not be captured by the current approach which looks at the overall norm.

A more concerning issue I see is the ablation in Table 6, where experiments are performed by only keeping the lowest scored features. While there is a significant regression in the medium cases, the difference for easy and hard cases is actually very minimal. The text describes this as a dramatic performance drop, but it's not convincing that this is actually the case. If the network is able to still achieve similar performance on hard examples when using what should be completely irrelevant features, what does that say about the foundational assumption that high norm features are the most important?

Also, in (4), the features are further weighted by the score for that feature. It's unclear why this is necessary, compared to only using the scores to select sites for convolutions. Does this weighting improve performance or provide some theoretical guarantees? Some ablations for this would be helpful. Otherwise, if the base assumption is not always correct, this weighting may serve to further remove the effect of some potentially important features.

This weighting also serves to provide a gradient on the feature norms (perhaps this is the intention?). This seems like it could be useful, but might also interfere with the final detection task. Some explanations/ablations would be useful.

The pruning ratio in the experiments is also not explained. It sounds like it is the top_k features, but could also be the score threshold.

L159: should this be absolute magnitudes instead of absolute means?

---

> ### Author Response · Authors · 2022-08-02
> **Response to B4Lc (2/2)**
>
>
> **AS2:  Performance ablation on SPRS-Conv**
>
> **(1) Experiments on SPRS-Conv on KITTI and Nuscens:** We performed inversion ablation experiments on SPRS-Conv to further validate our proposed hypothesis. Similar to table 6 in main paper, we exchanged the important and unimportant area which means only the positions with low magnitude would be dilated. The experiments are conducted on both KITTI and nuScenes dataset. We report the results as below:
>
> | Method (KITTI)    | Easy  | Moderate | Hard  |
> | ----------------- | ----- | -------- | ----- |
> | SPRS-Conv         | 89.22 | 84.36    | 78.83 |
> | SPRS-Conv inverse | 70.36 | 49.81    | 44.06 |
>
> | Method(nuScenes)  | mAP   | NDS   | car  | truck | bus  | trailer | construction_vehicle | pedestrian | motorcycle | bicycle | traffic_cone | barrier |
> | ----------------- | ----- | ----- | ---- | ----- | ---- | ------- | -------------------- | ---------- | ---------- | ------- | ------------ | ------- |
> | SPRS-Conv         | 58.48 | 66.11 | 85.0 | 58.2  | 69.5 | 35.7    | 15.5                 | 85.3       | 58.8       | 40.9    | 70.0         | 68.1    |
> | SPRS-Conv inverse | 16.72 | 39.29 | 29.6 | 9.4   | 16.3 | 1.7     | 0.6                  | 24.2       | 11.8       | 3.6     | 29.0         | 41.1    |
>
> **(2) Analysis of Experimental Results**: Compared with the experimental results of SPSS-Conv, the inversion experiment of SPRS has a more exaggerated performance loss. This experimental phenomenon is also easy to understand. When inversion is not performed, SPRS will select a position with a large magnitude for expansion, and the rest will be suppressed.  We believe that it is beneficial to the task, and the irrelevant features are reduced in the downsample part, which shows that there is no obvious performance loss in the result. However, when we choose to suppress positions with large magnitudes, the spatial redundancy is amplified, and important features cannot be effectively expanded, and even discarded due to downsampling. After the above features are converted to Bird's Eye View, since the number of foreground points from the input is weakened, the effective features extracted by BEVbackone are very limited, resulting in a great degree of performance degradation. We will add and discuss these results in the final version of the paper.
>
> **Q3: Why it is necessary to multiply the feature with the magnitude mask?**
>
> Thank you for this great comment.
>
> **A3:  (1) why multiplying magnitude mask:** For this problem, our initial purpose is to let the magnitude mask as a bridge to provide additional gradients for supervising the feature norm, further enhancing the difference between important and non-important features. As the network is end-to-end optimized for the object detection task, the additional gradient will not interfere with the original gradient but instead try to make areas that are important for detection have a larger magnitude.
>
> **(2) Necessity of the multiplication operation:** We do further investigation on whether this multiplication is necessary. We observe that it only brings marginal performance gains as shown in Table below. This further confirms that without any additional guidance, the magnitude of features from a detection network is sufficient to serve as a good criterion for deciding important vs unimportant regions. This strengthens our initial claim and echoes our motivation of using magnitude as a selection criterion.
>
> | Method (KITTI)           | Easy  | Moderate | Hard  |
> | ------------------------ | ----- | -------- | ----- |
> | SPSS-Conv                | 89.22 | 84.36    | 78.83 |
> | SPSS-Conv (not multiply) | 89.02 | 84.13    | 78.81 |
>
> | Method (nuScenes)        | mAP   | NDS   |
> | ------------------------ | ----- | ----- |
> | SPSS-Conv                | 58.48 | 66.11 |
> | SPSS-Conv (not multiply) | 58.27 | 66.01 |
>
> (3) We will make this clear in the final version.
>
> **Q4: The pruning ratio in the experiments is also not explained？**
>
> **A4:** The way of dynamic division in SPS-Conv is optional, such as using a fixed threshold or simply taking the elements with top-k scores.  During our experiments, in order to better control the pruning ratio, we choose the top-k result as the indicator. Sorry for the confusion here, we will correct and note in the article.
>
> **Q5: L159: should this be absolute magnitudes instead of absolute means?**
>
> **A5:** Thanks for the suggestion. We will correct it to “calculate the channel-wise absolute mean values on different voxels”.

---

> ### Author Response · Authors · 2022-08-02
> **Response to B4Lc (1/2)**
>
> **Q1:  Large feature magnitude vs high feature absolute value in a channel**
>
> **A1:**  Thanks for this insightful suggestion. We are very sorry that this question was overlooked when designing the experiment. According to the suggestion, we have the following views on this issue:
>
> **(1)** **Large overlap in selected points:** Here, we use channel-wise absolute mean (feature l_1 norm) and absolute max to select important positions and calculate their intersection portions. The experimental results show that the candidate sets selected by the two methods have an intersection rate of more than 87%. Therefore, we have reason to believe that there is a certain consistency of results between the two criteria because the samples whose average feature norm is small but feature norm on individual channels is large are minorities.
>
> **(2) Performance analysis:** Since our approach of using absolute mean to obtain magnitude has already achieved similar performance as the baseline, we think that even adding those outliers (those features with very large values on some specific channels) will not increase the performance any further.
>
> **Q2:  Ablation results in table 6.**
>
> **A2:**  Thanks for the suggestion. After rethinking table 6, we found that it may indeed bring some confusion, so we will further prove the effectiveness of our method from the following two aspects.
>
> 1. Why is the performance drop limited in easy and hard cases?
> 2. What is the effect of the inverse experiment on SPRS-Conv?
>
> **AS1: Performace drop analysis on SPSS-Conv**
>
> **(1) Background of the detection architecture:** The entire 3D detector contains two feature extraction parts: sparse CNN and BEVbackone. For sparse CNN, it takes a 3D point cloud as input and extracts spatial geometric information. In turn, the height information of these features will be compressed into the channel dimension, and the point cloud will be converted into a regular 2D feature. BEVbackbone, as a feature encoder composed of only 2D convolutions, can further obtain high-quality feature representation. Since the two feature extractors are coupled together, they are both indispensable for bounding box regression.
>
> **(2) Explanation of Table 6 on the KITTI dataset**: For convolution in unimportant positions in table 6, the easy and hard cases do not suffer from performance degradations. We think the most important cause for this phenomenon is the KITTI dataset which is a small-scale dataset and potentially has an in-balanced or skewed distribution of objects in different ranges.  This makes the sparse CNN component more crucial to the moderate objects while less crucial for easy and hard cases. In order to verify this conjecture, we removed all the submanifold layers in the sparse CNN, leaving only the downsample layer to align the shape of the feature, the result is shown in the table below.
>
> | Method (KITTI)        | Easy  | Moderate | Hard  |
> | --------------------- | ----- | -------- | ----- |
> | SPSS-Conv             | 89.22 | 84.36    | 78.83 |
> | SPSS-Conv inverse     | 89.15 | 79.13    | 78.47 |
> | basline no subm layer | 88.84 | 78.88    | 78.25 |
>
> Surprisingly, compared to the baseline, even if 3D convolution is not used for feature extraction, the performance of easy and hard will not fluctuate significantly. Therefore, we have reason to suspect that sparse CNN itself is redundant for these easy and hard objects on the KITTI dataset, which is caused by the characteristics of the data itself.
>
> **(3) Experiments on the nuScenes dataset:** For further evaluation, we replicate this ablation study on a much larger dataset — nuScenes to verify our claim. The results are shown in the following table:
>
> | Method            | mAP   | NDS   | car  | truck | bus  | trailer | construction_vehicle | pedestrian | motorcycle | bicycle | traffic_cone | barrier |
> | ----------------- | ----- | ----- | ---- | ----- | ---- | ------- | -------------------- | ---------- | ---------- | ------- | ------------ | ------- |
> | SPSS-Conv         | 58.48 | 66.11 | 85.0 | 58.2  | 69.5 | 35.7    | 15.5                 | 85.3       | 58.8       | 40.9    | 70.0         | 68.1    |
> | SPSS-Conv inverse | 55.84 | 64.72 | 84.3 | 56.7  | 68.2 | 33.3    | 14.4                 | 83.5       | 54.4       | 34.2    | 63.3         | 66.1    |
>
> Based on the results in the above table, it can be seen that when we reverse the important position, there is a considerable performance drop in all categories. This phenomenon is more obvious in small objects, especially for the traffic cone and bicycle categories (even reaching about a 7% performance drop). The results differences are caused by the difference in terms of size and point numbers. Large object categories (such as vehicles) with more points are less sensitive to the sampling method but still have a notable performance drop. We will add the ablation results in the paper.

---

### Official Review · Reviewer_cuTj · 2022-07-11

**Rating:** 5
**Confidence:** 4
**Soundness:** 3 good
**Presentation:** 3 good
**Contribution:** 2 fair

**Summary:**

This paper studies the spatial redundancy in LiDAR-based 3D object detection. The authors propose spatial pruned sparse convolution (SPS-Conv) that skips computation for voxels with lower activation values. The authors have evaluated their proposed method on KITTI and nuScenes, achieving a 50% reduction in #MACs without loss of accuracy.

**Questions:**

I would love to see the answer to the following questions in the rebuttal:
* What is the latency reduction achieved with the proposed SPS-Conv?
* How does random voxel dropout work compared with magnitude-based pruning?
* How does the proposed SPS-Conv work on Waymo with much denser point clouds?

The authors could refer to the previous section for more detailed comments.


**Limitations:**

The authors have addressed the limitations and potential negative societal impact of their work.


**Strengths And Weaknesses:**

**[Strengths]**

The paper is well-written and easy to follow. The proposed SPS-Conv is well-motivated, technically sound and achieves good empirical performance on two large-scale benchmarks. The authors have provided sufficient implementation details, which could facilitate the reproduction. The authors have also promised to release the code upon acceptance.

**[Weaknesses]**

The authors use #GFLOPs as the primary efficiency metric throughout the paper. However, the reduction in #MACs does not necessarily translate into the measured speedup on the actual hardware (due to other costs such as data movement). I would highly recommend the authors report the measured latency on GPUs. As 3D object detectors contain more than sparse convolutions (since there are extra computations in the BEV decoder), it is completely acceptable to just present the latency of the sparse LiDAR encoder.

The inference engine has a substantial impact to the final inference speed. It would be great if the authors could try out the three available sparse convolution inference engines, including SpConv, TorchSparse and MinkowskiEngine, to see whether the proposed method can achieve consistent speedup with all of them.

The authors claim that the activation magnitude serves as a good indicator of selecting important voxels. It would be more interesting to see how random voxel dropout works (in additional to comparing with the inverse magnitude baseline).

Finally, the authors have evaluated their proposed method on KITTI and nuScenes. It would be great if the authors could also evaluate their method on Waymo since the point cloud data on Waymo is much denser (and could be more redundant).

---

> ### Author Response · Authors · 2022-08-02
> **Response to Reviewer cuTj**
>
> **Q1：The latency reduction of the SPS-Conv ?**
>
> **A1**:  Please check out Q1 of the common question section, where we provide detailed latency test results. In the table, we quantitatively present the contributions of individual components of our proposed method. Due to the difference in the densities of the point clouds of the two datasets and the numbers of convolutional layers of the sparse CNNs, a discrepancy on the acceleration effects is inevitable.
>
> The results in Table (common question Q1)  show that using SPSS-Conv or SPSS-Conv alone can achieve more than 10% speed improvement. When they are used in combination, there is a speed increase of around 20%. This means that our method can deliver a notable reduction in latency. Note that we just use standard Pytorch and CUDA combined programming and do not optimize the implementation, which indicates that there still exists room for the further latency reduction as latency is very relevant to implementations. We will add this to our paper.
>
> **Q2：Inference speed on different inference engines ?**
>
> **A2**:  Thanks for the suggestion. In fact, when we started to build this project, we did consider the choice of the inference engine. As the codebase of spconv is more extensible, our first version of the code was built on spconv. Due to the limited time in the rebuttal period, we cannot complete the development of the other two inference engines in such a short time. We are committed to further porting our code into their codebase and reporting the results in our final version.
>
> **Q3：How does random voxel dropout work compared with magnitude-based pruning ?**
>
> Thank you for this insightful question. We conducted experiments and analysis as below.
>
> **A3:   Experiments on random voxel dropout:** We carried out the experiment of random drop ablation on both KITTI and nuScenes datasets. The ratio of random drop is set the same as our magnitude-based pruning:   for KITTI, we set pruning ratios in SPSS-Conv and SPRS-Conv as 0.5 and 0.5 respectively; as for nuScenes, they are set as 0.3 and 0.5.  The table below shows the performance comparison of random drop and magnitude as indicators.
>
> | Method (nuScenes)     | mAP   | NDS   |
> | --------------------- | ----- | ----- |
> | SPSS-Conv             | 58.48 | 66.11 |
> | SPSS-Conv inverse     | 55.84 | 64.72 |
> | SPSS-Conv random drop | 56.12 | 64.49 |
> | SPRS-Conv             | 58.59 | 66.23 |
> | SPRS-Conv inverse     | 16.72 | 39.29 |
> | SPRS-Conv random drop | 55.58 | 64.34 |
>
> | Method (KITTI)        | Easy  | Moderate | Hard  |
> | --------------------- | ----- | -------- | ----- |
> | SPSS-Conv             | 89.22 | 84.36    | 78.83 |
> | SPSS-Conv inverse     | 89.15 | 79.13    | 78.47 |
> | SPSS-Conv random drop | 89.14 | 83.21    | 78.57 |
> | SPRS-Conv             | 89.64 | 84.26    | 78.91 |
> | SPRS-Conv inverse     | 70.36 | 49.81    | 44.06 |
> | SPRS-Conv random drop | 89.32 | 78.81    | 78.28 |
>
> **(1) Magnitude-base pruning vs Random drop**: compared to magnitude-based pruning, we observe using random drop as an indicator will lead to a certain loss in performance (around 2%)。This is caused by the randomness, part of the foreground is discarded, resulting performance degradation. However, the important part still has a 50% chance of being selected, which also guarantees performance to a certain extent.
>
> **(2) Analysis on random drop**: Randomly dropping points obtain reasonable results on both datasets. This further confirms our observation about the extreme imbalance of foreground and background. Even randomly dropping points, we still have a reasonable chance of dropping useless points.
>
> **(3) Drawback of random drop**: Despite of its degraded performance, the random drop method also has a certain degree of randomness. This is not desirable in practical applications as it may have the chance to lose some safety-critical areas which will cause problems in safety-critical applications.
>
> **Q4：How does the proposed SPS-Conv work on Waymo with much denser point cloud?**
>
> **A4:**  We show the results of the proposed method on the Waymo dataset in Q2 of the Common question. As shown in the table, our method is also able to maintain competitive performance on various metrics on this dataset while saving 63% GFLOPs. This further illustrates the generality of our method. We will add and discuss these results in the revised version of the paper.

---

> > ### Comment · Reviewer_cuTj · 2022-08-09
> > **Response**
> >
> > Thanks for your reply which resolves most of my concerns! I will keep my original rating.

---

> > > ### Author Response · Authors · 2022-08-10
> > > **Author Response**
> > >
> > > We thank the reviewer again for the detailed discussions and the kind support of this work. Your constructive feedback and criticisms will help us greatly towards improving this work.

---

### Official Review · Reviewer_TzZm · 2022-07-12

**Rating:** 5
**Confidence:** 3
**Soundness:** 2 fair
**Presentation:** 2 fair
**Contribution:** 3 good

**Summary:**

This paper proposed a non-learning convolution operator that simply use magnitude of feature as cue to remove redundant background points in the sparse convolution operation. The authors validated the proposed operator by combining it to popular sparse convolutional network based object detection and achieve more than 50% reduction in GFLOPs without compromising the performance on benchmark datasets.

**Questions:**

Is it possible that the effectiveness of the propose method comes from the fact that background points has been filtered out by magnitude, thanks to the characteristics of the input point cloud feature representation used in the experiment?

**Limitations:**

I suggest the author to add visualization point cloud that belongs to important set to help with understanding.

**Strengths And Weaknesses:**

Strengths:
- Simplicity of the proposed method

Weaknesses
- It is not clearly stated why magnitude is the key to discriminate foreground and background points. In the experimental section, the authors combines the operator with existing object detection networks CenterPoint and Voxel R-CNN. But the effectiveness of the methods seems to be more related to feature/representation of the input point-cloud and the authors didn’t discuss that part in the paper.

---

> ### Author Response · Authors · 2022-08-02
> **Response to Reviewer TzZm**
>
> **Q1:   It is not clearly stated why magnitude is the key to discriminate foreground and background points？**
>
> Thank you for the comment.
>
> **Visual Analysis：** To better understand points pruned by our magnitude criterion, we visualize point clouds before and after pruning. Note that the point clouds used for visualization are randomly chosen from the nuScenes dataset. The comparison results are shown in the link [[visual](https://drive.google.com/drive/folders/1aoQOrYRB57tKGHymMg3IuS2DRoLh00wR?usp=sharing)], we provide the original image and the pruned image with the file names _raw.png and _im.png respectively. And we roughly annotate the positions of cars (red) and pedestrians (yellow). we observe that most of the foreground points are preserved. For the background areas, points that fall in vertical structures, such as light, poles, and trees, are also preserved  as they tend to be hard negatives, and easily confused with foreground objects. These points require a deep neural network with a certain capability to process in order to recognize them as background.  In contrast, background points in flat structures such as road points are largely removed because they are easily identifiable redundant points.
>
> **Why foreground points with high feature magnitude?**  To gain more insights into why high feature magnitude corresponds to the above patterns, we conjecture that this is caused by the training objective in 3D object detection. When training a 3D object detection model, the focal loss is adopted as default in 3D object detection. When we look closer at the focal loss, it will incur a loss on positive samples and hard negatives while easy negatives are removed from the loss. Thus, this will generate gradients in the direction that can incur an update of features for areas with positive samples and hard negatives. This can eventually make a difference in their feature magnitudes in comparison with areas for easy negatives which are less frequently considered in the optimization objective.
>
> We will add the above to our paper.
>
> **Q2:   Feature representations in the point cloud.**
>
> Thanks for the comment. We hope our explanation in Q1 resolves your concern regarding our selection criterion. As we remove redundant points in intermediate layers, this indeed will have an impact on feature representation learning as the topology of point clouds might change. But as our performance doesn’t drop, this change of topology at least is not harmful to model performance and is effective in maintaining the original capability of our model. This in turn reflects that our selection criterion is successful which removes points but can still maintain model effectiveness. Also, since the model is optimized end-to-end, representation learning and spatial pruning based on magnitude are integrated together as a whole, it is difficult to quantify the contribution of each one solely. We will add this to our paper.

---

> > ### Comment · Reviewer_TzZm · 2022-08-10
> > **Response**
> >
> > Thanks for the authors for providing the visualization and add the analysis for the relationship between magnitude and visualization to address my concern. While the methodology makes sense with those added analysis, I feel the visualization could still be further explored to understand better how feature magnitude varies in the training process before pruning. I will suggest to use heat map visualization for the feature magnitude. Despite the flaws in the representation, I do think this paper provided valuable and practical contribution to the field. I will changed my overall rating of this paper to be: 5: Borderline accept.

---

> > > ### Author Response · Authors · 2022-08-10
> > > **Response to Reviewer TzZm**
> > >
> > > Dear reviewer TzZm:
> > >
> > > We are sincerely grateful for your positive feedback of our work. Thanks for your remind that the visualization should still be further explored. We are preparing and will add these to our paper.
> > >
> > > Best regards,
> > >
> > > Paper 1506 authors

---

> ### Author Response · Authors · 2022-08-09
> **Feedback and Discussion**
>
> Dear reviewer TzZm:
>
> Thanks for your great efforts and insightful suggestions. We have provided responses to your concerns. If you still have other questions, we are happy to further discuss with you. Thanks for your time again.
>
> Best regards,
>
> Paper 1506 authors

---

### Author Response · Authors · 2022-08-02
**Response to All Reviewers**

Dear all reviewers:

We appreciate all the reviewers' valuable time and suggestions. In this section, we will first give detailed answers to some common questions, followed by responses to each reviewer separately.

**Q1: The latency reduction achieved with the proposed SPS-Conv (Reviewer-cutj/Reviewer-aQxF)**

Thanks for the insightful comments. We conduct experiments and perform an analysis on this aspect as below.

(1) Latency reduction experiments: Here we also provide the latency reduction by our approach as shown in the following table.  We evaluate the effects of SPSS-Conv and SPRS-Conv alone and their combination on the KITTI and nuScenes datasets. All experiments are tested on the same server, using a single 2080ti GPU with the batch size set to 1.  We also address specific questions/concerns regarding this study. Please refer to our separate response to each reviewer.

| Method / speed(ms) | KITTI (VoxelNet) | speed up | GFLOPs | nuScenes (VoxelResNet) | speed up | GFLOPs |
| :----------------: | ---------------- | -------- | ------ | ---------------------- | -------- | ------ |
|      baseline      | 40ms             | None     | 7.6    | 50ms                   | None     | 62.9   |
|        spss        | 36ms             | 10%      | 4.51   | 44ms                   | 12%      | 45.2   |
|        sprs        | 33ms             | 17.5%    | 4.24   | 44ms                   | 12%      | 47.2   |
|     sprs+spss      | 30ms             | 25%      | 3.6    | 41ms                   | 18%      | 34.3   |

(2) **Latency vs FLOPs:** As shown in the table above, the reduced FLOPs cannot all translate into a reduction in latency but we still obtain a notable reduction in latency (KITTI 25%, nuScenes 18%) without sacrificing accuracies.  This is because, besides the FLOPs of the model itself, latency is also closely related to hardware-level implementations and optimizations. Note that our model is implemented using PyTorch functions without specific optimization. Better hardware-level implementation of operations has the potential to reduce latency, which is beyond the investigation of this paper and will be our future work.

(3) **Importance of FLOPs in energy efficiency:** Besides, given the same hardware, the number of FLOPs will determine the amount of energy consumption [1] at the inference stage: FLOPs measure the total number of operations required to compute the output, and each operation will cost certain energy depending on the hardware. Therefore, the large reduction in FLOPs also has positive impacts on practical applications in power-constrained scenarios.

1. Desislavov, Radosvet, Fernando Martínez-Plumed, and José Hernández-Orallo. "Compute and energy consumption trends in deep learning inference." *arXiv preprint arXiv:2109.05472* (2021).

(4) We will add all the above in our paper.

**Q2: Experimental results on Waymo dataset (Reviewer-cutj/Reviewer-aQxF)**

Thank you for the suggestion to help improve our paper.  In accordance with the requirements of R2 and R4, we conduct experiments on Waymo, a dataset with a larger amount of dense point cloud data.

(1) **Experimental setting:** Due to the limited time for rebuttal, we follow the setting in OpnePCDet and conduct experiments on one-fifth of the sub-data set which is an standard setting and doesn’t sacrifice the performance much.  We choose a strong method, CenterPoint, as our baseline. The experimental results are shown below.

| Performance@(train with 20% Data) | GFLOPs | Vec_L1      | Vec_L2      | Ped_L1      | Ped_L2      | Cyc_L1      | Cyc_L2      |
| --------------------------------- | ------ | ----------- | ----------- | ----------- | ----------- | ----------- | ----------- |
| CenterPoint (ResNet)              | 76.7   | 72.76/72.23 | 64.91/64.42 | 74.19/67.96 | 66.03/60.34 | 71.04/69.79 | 68.49/67.28 |
| CenterPoint + SPSS                | 43.5   | 72.46/71.91 | 64.35/63.85 | 73.71/67.53 | 65.81/60.13 | 70.85/69.63 | 68.52/67.27 |
| CenterPoint + SPRS                | 55.2   | 72.80/71.96 | 64.36/64.07 | 73.99/67.84 | 66.05/60.40 | 70.42/69.69 | 68.85/67.37 |
| CenterPoint + SPSS + SPRS         | 28.8   | 72.68/72.13 | 64.66/64.16 | 74.39/68.02 | 66.00/60.36 | 71.08/69.84 | 68.47/67.28 |

(2) **Analysis of results:**

The above experimental results demonstrate the generality of SPS-Conv on large-scale datasets with dense point clouds: SPS-conv still maintains the high performance while significantly reduces FLOPs. We will also include the above results in the revised version of the paper.



#### Other Concerns

Other concerns regarding typo, clarity and figures have been properly addressed in the revised paper.

---

### Author Response · Authors · 2022-08-07
**Looking forward to the reviewer's reply**

Dear reviewers,

Thanks a lot for your time and efforts in reviewing our paper. We have tried our best to address all mentioned concerns. We would appreciate it if you can take a look at our response. Your feedback is valuable to us, and we are willing to have further discussions with you.

Best regards,

Paper 1506 Authors

---

### Meta-Review · Area_Chair_AyPm · 2022-08-26

**Recommendation:** Accept
**Confidence:** Certain

**Metareview:**

The paper shows that it is possible to obtain a good saving in both terms of FLOPS and latency using sparse convolutions for 3d object detection by leveraging the magnitude of features. After a strong rebuttal all 4 reviewers vote for acceptance of the paper with high-confidence.

I suggest that the authors incorporate the comments from reviewers and some of the results from the rebuttal to make the paper more immediately convincing upon a first reading.

**Award:**

No

---

### Decision · Program_Chairs · 2022-09-14

Accept